# On-Slide Lambda Protein Phosphatase-Mediated Dephosphorylation of Fixed Samples

**DOI:** 10.3390/mps6030055

**Published:** 2023-05-27

**Authors:** Alexander Tishchenko, Cliff Van Waesberghe, Herman W. Favoreel

**Affiliations:** Department of Translational Physiology, Infectiology and Public Health, Faculty of Veterinary Medicine, Ghent University, 9820 Merelbeke, Belgium; alexander.tishchenko@ugent.be (A.T.);

**Keywords:** protein phosphorylation, lambda phosphatase, dephosphorylation assay, immunofluorescence, phospho-specific antibodies

## Abstract

Protein phosphorylation is a ubiquitous post-translational modification that regulates a plethora of intracellular processes, making its analysis crucial for understanding intracellular dynamics. The commonly used methods, such as radioactive labeling and gel electrophoresis, do not provide information about subcellular localization. Immunofluorescence using phospho-specific antibodies and subsequent analysis via microscopy allows researchers to assess subcellular localization, but it typically lacks validation whether the observed fluorescent signal is phosphorylation specific. In this study, an on-slide dephosphorylation assay coupled with immunofluorescence staining using phospho-specific antibodies on fixed samples is proposed as a fast and simple approach to validate phosphorylated proteins in their native subcellular context. The assay was validated using antibodies against two different phosphorylated target proteins, connexin 43 phosphorylated at serine 373, and phosphorylated substrates of protein kinase A, with a dramatic reduction in the signal upon dephosphorylation. The proposed approach provides a convenient way to validate phosphorylated proteins without the need for additional sample preparation steps, reducing the time and effort required for analysis, while minimizing the risk of protein loss or alteration.

## 1. Introduction

Protein phosphorylation is the most common type of post-translational modification. This process is reversible and involves the transfer of a gamma-phosphate group from ATP to a specific amino acid residue (in eukaryotes, this is usually serine, threonine, or tyrosine) via the action of protein kinases [1]. The addition of a phosphate group may lead to changes in activity, stability and/or localization of proteins, and in its interaction with other proteins. Protein phosphorylation events regulate a plethora of key processes in intracellular dynamics, including cell growth, differentiation, metabolism, enzymatic activity, and signal transduction [2]. The role of protein phosphorylation as the main post-translational regulatory mechanism in eukaryotic proteins can at least partly be attributed to its simple and highly dynamic nature, in combination with the broad availability of ATP as a phosphoryl group donor.

It is estimated that the mammalian proteome could contain as much as 10^5^ phosphosites [3,4,5]. In addition, it has been estimated that at least a third of the mammalian proteins can be phosphorylated [6]. Meanwhile, genome sequencing has revealed that 2–3% of all eukaryotic genes may encode protein kinases [7,8].

Mass spectrometry has emerged as a powerful tool and an unbiased method for protein phosphorylation analysis [9]. However, this approach comes with several limitations, such as the loss of phosphates during sample preparation, equipment costs, or the fact that many phosphoproteins are physiologically expressed in low abundance [10,11,12].

Other procedures that are commonly used to identify phosphorylated proteins include radioactive labelling using 32P-labeled ATP, followed by gel electrophoresis, gel electrophoresis in combination with phosphospecific antibodies, or evaluating the changes in apparent mobility upon gel electrophoresis after a dephosphosphorylation assay. In all these assays, the removal/reversal of the phospho-specific protein band via the incubation of whole cell lysates or immunoprecipitated protein with lambda protein phosphatase serves as an important control [13]. However, these types of assays and approaches do not provide information regarding the potential differences in phosphorylated protein abundance in individual cells and/or the subcellular localization of the phosphorylated protein, which might be key information depending on the type of research. Immunofluorescence assays using phospho-specific antibodies followed by microscopic analysis allow researchers to retrieve this type of information, but they typically lack controls to confirm that the observed signal is phospho-specific.

In the current study, we describe an immunofluorescence-based protocol to confirm protein phosphorylation using phospho-specific antibodies. The assay consists of the on-slide dephosphorylation of fixed cells using lambda protein phosphatase, followed by a standard immunofluorescence protocol using specific antibodies against phosphorylated proteins. This approach validates the visualization of the target phospho-protein in its cellular context.

## 2. Materials and Methods

### 2.1. Cell Culture

Swine testicle epithelial (ST) cells (ATCC CRL-1746; Sus scrofa, pig) were cultured in Modified Eagle’s Medium (MEM) supplemented with 10% inactivated fetal bovine serum (FBS), 100 U/mL penicillin, 0.1 mg/mL streptomycin, 50 μg/mL gentamicin, and 1 mM sodium pyruvate (all from Gibco, Thermo Fisher Scientific, Waltham, MA, USA).

Swine kidney epithelial (SK-6) cells (RRID: CVCL_D296; Sus scrofa, pig) were cultured in MEM supplemented with 10% inactivated FBS, 100 U/mL penicillin, 0.1 mg/mL streptomycin, 50 μg/mL gentamicin, and 1 mM sodium pyruvate.

ST and SK cells were grown to confluence on glass coverslips in a 24-well plate and fixed using ice-cold methanol for 10 min at room temperature.


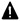
 **CRITICAL STEP**: Methanol fixation is the most suitable option since it mostly maintains protein antigenicity intact.

### 2.2. On-Slide Dephosphorylation Protocol

The fixed cells were further permeabilized for 10 min at room temperature with 0.1% Triton X-100 dissolved in Dulbecco’s Phosphate Buffered Saline solution (DPBS) containing CaCl_2_ and MgCl_2_ (Thermo Fisher Scientific, Cat. no.: 14040091). Later, dephosphorylation was carried out using a solution containing: 3 μL of lambda protein phosphatase (PPse) (400,000 units/mL) (New England Biolabs, Ipswich, MA, USA, Cat. no.: P0753S, component #P0753SVIAL), 15 μL NEBuffer Pack for Protein MetalloPhosphatases (PMP) (New England Biolabs, Cat. no.: P0753S, component #B0761SVIAL), 15 μL MnCl_2_ (New England Biolabs, Cat. no.: P0753S, component #B1761SVIAL), and 467 μL of DPBS (CaCl_2_ and MgCl_2_). A total of 500 μL of lambda PPse-containing solution were used per slide. As a control, a slide was incubated with the digestion mix (without lambda PPse) containing 15 μL of PMP buffer and 15 μL of MnCl_2_ dissolved in 470 μL of DPBS (CaCl_2_ and MgCl_2_). As an additional negative control, one slide was incubated with a solution containing: 3 μL of lambda PPse, 15 μL of PMP Buffer, 15 μL MnCl_2_, 5 μL of 0.5 M EDTA at pH 8.0 (Invitrogen, Waltham, MA, USA, Cat. no.: AM9260G), and 462 μL of DPBS (CaCl_2_ and MgCl_2_). The slides were incubated for 3 h at 30 °C. A detailed workflow diagram for the on-slide dephosphorylation assay is shown in Figure 1. 

### 2.3. Immuno-Staining

Following lambda PPse digestion (or the same treatment without the addition of PPse), the slides were rinsed once with 1 mL DPBS (CaCl_2_ and MgCl_2_), and then blocked using a solution of 3% (*w*/*v*) bovine serum albumin (BSA) and 0.1% Triton X-100 diluted in DPBS (CaCl_2_ and MgCl_2_) for 1h at 37 °C. The slides were rinsed again with 1 mL DPBS (CaCl_2_ and MgCl_2_) and incubated with the primary antibodies overnight at 4 °C.

The following primary antibodies were diluted in DPBS (CaCl_2_ and MgCl_2_) containing 1% (*w*/*v*) BSA at the following concentrations: rabbit mAb anti-phospho PKA substrate (1:300 dilution; Cell Signalling Technology, Danvers, MA, USA, Cat. no.: 9624) and rabbit pAb anti-phospho-Connexin 43 (Ser373) (1:200 dilution; Invitrogen, Cat. no.: PA5-64670). After overnight incubation, the unbound primary antibody was removed via three washing steps with DPBS (CaCl_2_ and MgCl_2_). Then, fluorochrome-conjugated goat anti-rabbit secondary antibody (1:200 dilution; Invitrogen) was added for 1 h at 37 °C. Cell nuclei were counterstained using Hoechst 33342 (1:200 dilution; Invitrogen, Cat. no.: H3570) for 10 min at room temperature.

### 2.4. Microscopy

Fluorescence images were taken using a Leica Thunder imaging system (Leica, Wetzlar, Germany) and were analyzed using ImageJ software (NIH, Bethesda, MD, USA).

## 3. Results and Discussion 

Several methods for assessing and validating phosphorylation have been developed, but most of them cannot determine the subcellular location of the targeted phosphoprotein. Detecting phosphorylated proteins via the microscopic analysis of immunofluorescence assays using phospho-specific antibodies achieves this, but it lacks validation as to whether the observed signal is phosphorylation-specific. In the current work, we present an on-slide lambda PPse dephosphorylation assay of methanol-fixed samples, coupled with a standard immunostaining protocol using phospho-specific antibodies, which overcomes this limitation. As shown in Figure 2b, dephosphorylation vastly reduced the immunofluorescent signal observed using a phospho-specific antibody that recognizes the gap junction protein, connexin 43 (Cx43), when it is phosphorylated at serine residue 373 (S373). The activity of lambda PPse requires divalent cations and is, therefore, inhibited via the addition of the divalent cation chelator EDTA [14]. Hence, as an additional control, 5 mM EDTA was added to the PPse digestion mix. As expected, the addition of EDTA blocked the catalytic activity of lambda PPse, and consequently, no reduction in the phospho-S373 Cx43 signal was observed (Figure 2c). These results indicate that the enzymatic dephosphorylation of S373 of Cx43 is indeed responsible for the decrease in fluorescence signal, rather than a possible allosteric change induced via the presence of lambda PPse.

Immunofluorescence staining using an antibody against phosphorylated substrates of protein kinase A (PKA) resulted in a predominantly nuclear localised signal, which is in line with earlier reports [15]. Following dephosphorylation, the signal reduction was still very substantial (Figure 3b), but less dramatic compared to that of phospho-Cx43 staining (Figure 2). We hypothesize that this difference in efficiency may be partly due to the difference in the subcellular localization of both types of signal. Cx43 is a transmembrane protein that may be relatively easily accessible for lambda PPse, whereas the signal observed using a phospho-PKA substrate-specific antibody was mainly localized in the nucleus, which is surrounded by a nuclear envelope and in close contact with chromatin, which possibly may hinder the access of lambda PPse to the phosphorylated residues, and thereby, reduce the efficiency of dephosphorylation.

To confirm that the obtained results are not cell-type-specific, in addition to the assays in ST cells, the current protocol was also applied to swine kidney (SK-6) cells stained against phospho-Cx43. SK-6 cells displayed lower basal levels of Cx43 phosphorylated at S373 compared to those of ST cells, but staining still yielded a clear signal that is mostly associated with the plasma membrane (Figure 4a). Moreover, the on-slide dephosphorylation protocol, again, virtually eliminated the phospho-Cx43 signal (Figure 4b). These results indicate that the current approach is robust and can be applied to cell lines derived from various origins.

It is worth mentioning that (phosphorylation-specific) antibodies may exhibit varying degrees of unspecific reactivity. This can sometimes pose challenges in interpreting the results. The current approach addresses this issue to some extent by assisting researchers in validating phospho-specific signals. However, with the current approach, in case there is a residual fluorescent signal following the dephosphorylation protocol, it is not possible to differentiate between incomplete dephosphorylation of the target phospho-antigen and unspecific antibody binding. In this regard, to further differentiate between both possible explanations, a comparison can be made between the fluorescent signal obtained when an antibody directed against the total target protein is used, irrespective of its phosphorylation, and the residual phospho-signal upon dephosphorylation. In case there is no overlap between both signals, this suggests that the residual fluorescent signal upon dephosphorylation is likely to be unspecific.

The proposed approach is simple, fast, uses materials commonly used in a molecular biology lab, and provides a convenient way to validate phosphorylated proteins in their native subcellular localization. The procedure can be performed on slides, without the need for additional sample preparation steps. This reduces the time and effort required for the analysis and minimizes the risk of losing or altering the protein of interest during the preparation steps. 

In future assays, it will be interesting to assess whether the current approach remains successful when it is applied to tissue sections. While the methanol fixation used in the current protocol can also be utilized for preparing immunohistochemistry samples in tissue sections, embedding processes may present challenges for successful tissue dephosphorylation. The commonly used paraffin wax and methylcellulose may reduce the ability of lambda PPse to access the phospho-antigens. Depending on the nature of the samples and the preparation protocol, an appropriate antigen retrieval step might be necessary to overcome this issue. Furthermore, tissue sections are often thicker and more complex than individual cells are. Depending on the tissue section thickness, lambda PPse would need to penetrate multiple layers of cells, the extracellular matrix, and/or the connective tissue to reach the target phospho-antigen.

## Figures and Tables

**Figure 1 mps-06-00055-f001:**
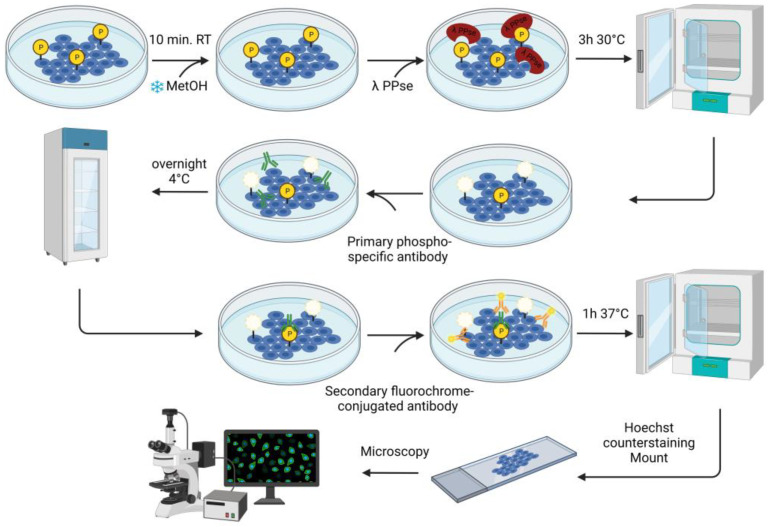
Schematic representation of the lambda protein phosphatase dephosphorylation protocol, coupled with a standard immunofluorescence protocol. As a control (not shown in the schematic representation), a slide with concurrently fixed cells was incubated with the exact same digestion mix without the addition of lambda PPse. Generated with BioRender.

**Figure 2 mps-06-00055-f002:**
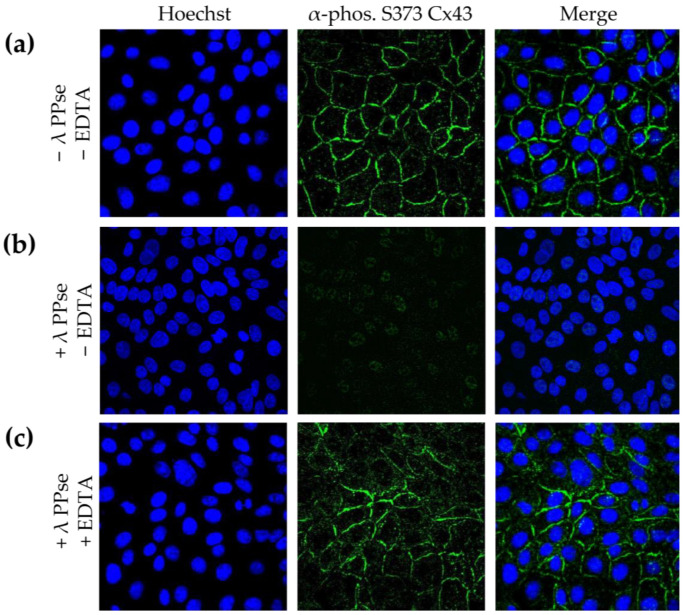
Immunofluorescence images of ST cells stained against phospho S373 Cx43. (**a**) Cells not treated with lambda PPse. (**b**) Cells treated with lambda PPse in the absence of EDTA. (**c**) Cells treated with lambda PPse in the presence of EDTA.

**Figure 3 mps-06-00055-f003:**
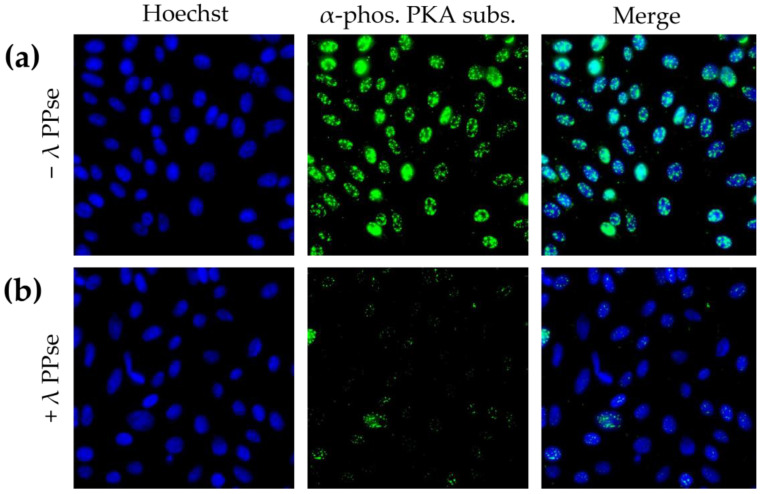
Immunofluorescence images of ST cells stained against phosphorylated substrates of PKA. (**a**) Cells not treated with lambda PPse. (**b**) Cells treated with lambda PPse.

**Figure 4 mps-06-00055-f004:**
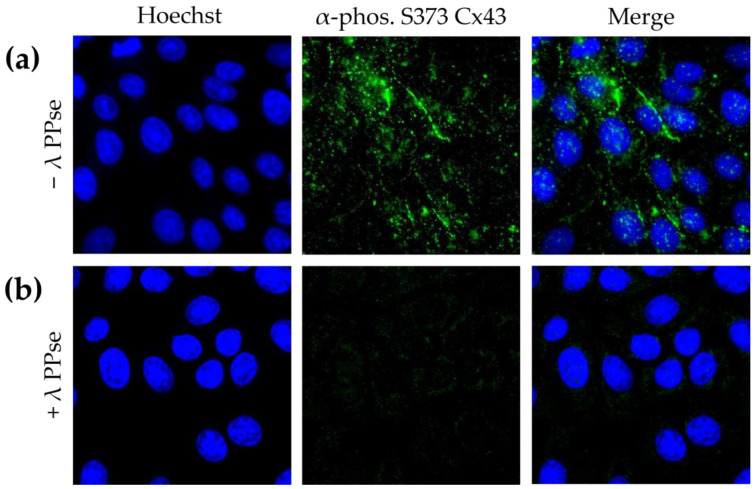
Immunofluorescence images of SK-6 cells stained against phospho S373 Cx43. (**a**) Cells not treated with lambda PPse. (**b**) Cells treated with lambda PPse.

## Data Availability

Not applicable.

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
