# Peer review of "On-Slide Lambda Protein Phosphatase-Mediated Dephosphorylation of Fixed Samples"

_mps, 2023, doi:10.3390/mps6030055_

Round 1

Reviewer 1 Report

The authors present an efficient, simple and inexpensive method to improve phosphoprotein research. The method is described completely and can be easily reproduced.

Only in part three (Results and discussion), some more details to the limitations of the methods should be given. The authors correctly mention that sometimes the lambda phosphatase possibly only partially dephosphorylate its targets, perhaps for various reasons.

But most commercially available antibodies, including phosphorylation-specific antibodies, show a lot of unspecific reactivity, leading to results in immunofluorescence staining that are difficult to interpret. The aim of the presented method is to avoid this issue. The authors hint at this problem at several points in the manuscript, but they should clearly mention that it is not possible by their method to differentiate between unspecific antibody binding and incomplete dephosphorylation.

Another limitation may be the use of only one cultured cell line for presentation of the method. Many researchers are also interested in phosphorylated proteins in tissue samples. Especially in tissue sections, unspecific antibody binding and/or incomplete dephosphorylation may occur. The authors should speculate on this theme and give a possible preview to future work testing their method with tissue sections.

Moreover, the authors should give hints to resolve this problem. For example, it is possible to compare the phosphorylation-specific immunofluorescence with a pan antibody staining. In the best case, the pan antibody and the phosphorylation-specific antibody were developed in different species, allowing double immunofluorescence staining. Then, there is a chance to differentiate between unspecific antibody binding and incomplete dephosphorylation, at least in part.

Minor: in line 59 there is a typo: it should be “phosphorylation” and not “phosphosphorylation”.
